# Case Report: Acute Intoxication from Phosphine Inhalation

**DOI:** 10.3390/ijerph20065021

**Published:** 2023-03-12

**Authors:** Longke Shi, Tianzi Jian, Yiming Tao, Yaqian Li, Guangcai Yu, Liwen Zhao, Zixin Wen, Baotian Kan, Xiangdong Jian

**Affiliations:** 1School of Public Health, Shandong University, Jinan 250012, China; 2Department of Poisoning and Occupational Diseases, Emergency Medicine, Qilu Hospital of Shandong University, Jinan 250012, China; 3School of Nursing and Rehabilitation, Cheeloo College of Medicine, Shandong University, Jinan 250012, China; 4Department of Geriatric Medicine, Department of Nursing, Qilu Hospital, Cheeloo College of Medicine, Shandong University, Jinan 250012, China

**Keywords:** phosphine, poisoning, inhalation, cardiac toxicity case report

## Abstract

Aluminum phosphide is a highly effective insecticide for fumigation in granaries and is often used in rural grain storage. However, people’s awareness of its toxicity is not strong. A case of acute inhalation toxicity of phosphine caused by the use of aluminum phosphide to fumigate a granary is reported here. The case presented with aspiration pneumonia and acute left heart failure. The patient was cured using comprehensive life support treatment, including respiratory support, antiarrhythmic treatment, and blood pressure maintenance with vasoactive drugs. There is no specific antidote for phosphine poisoning at present, and the comprehensive application of restricted fluid resuscitation, high-dose glucocorticoid shock, vasoactive drugs and bedside hemofiltration is significant in improving the prognosis of patients. It is also important to remind people to pay attention to their own protection in the process of using aluminum phosphide.

## 1. Introduction

Aluminum phosphide (AP) is a highly toxic insecticide (fumigant) that decomposes to generate a toxic phosphine gas in water or humid air, during storage and transportation, with use, as well as with exposure to rain and the sun. AP poisoning directly affects the cardiovascular system, and may lead to multiple organ system failure [1,2]. It causes noncompetitive inhibition of mitochondrial cytochrome C oxidase, electron transport chains, and oxidative phosphorylation. Inhibition of energy production in mitochondria leads to toxicity, extensive cell death and damage, and necrosis of the gastrointestinal tract, liver, and kidney [3]. On 27 July 2018, a case of acute poisoning caused by inhalation of phosphine during fumigation of a granary was admitted to Qilu Hospital of Shandong University (Lixia District, Jinan City, Shandong Province). After comprehensive life support treatment, including respiratory support, anti-arrhythmia, and vasoactive drugs to maintain blood pressure, the patient was cured.

## 2. Case Description

On 22 July, a 15-year-old male patient, approximately 173 cm in height and 80 kg in weight, went to a local hospital for treatment with nausea and chest tightness as the main symptoms. Routine blood tests, liver and kidney function tests, and an electrocardiogram were performed. The patient underwent no special treatment and was sent back home. On 23 July, while performing fumigation of the granary, the patient suddenly lost consciousness and was foaming at the mouth. He was taken back to the local hospital where he entered a coma and received endotracheal intubation as well as ventilator-assisted respiration. The local hospital administered IV fluids with diuretics for rehydration and to prevent fluid overload, respectively. Further imaging using bedside chest radiography (Figure 1) and an electrocardiogram were performed. Thereafter, the patient was transferred to the intensive care unit at 18:45 for intermittent hemoperfusion. Epinephrine and norepinephrine were continuously administered, and blood pressure was maintained at 120/70 mmHg. On 25 July, results from the local hospital showed a glutamic pyruvic transaminase level of 2934 U/L; glutamic oxaloacetic transaminase, 1870 U/L; total bilirubin, 30.7 μmol/L; and indirect bilirubin, 25.7 μmol/L. Further, on 26 July, the glutamic pyruvic transaminase level was 2536 U/L and the glutamic oxaloacetic transaminase level was 857 U/L. For further treatment, the patient’s family members transferred him to our hospital. The patient was located at Bozhou People’s Hospital in Anhui Province, which is more than 400 km away from us. After more than 4 h of ambulance transport, the patient arrived at our hospital at 9:00 a.m. on 27 July, and still required continuous instrumental ventilation.

The patient’s physical examination on admission was as follows: his body temperature was 36.8 °C; heart rate, 142 beats/min; respiratory rate, 28 times/min; and blood pressure, 103/60 mmHg (60 µg/min of norepinephrine). While in a sedated state (50 mg midazolam was added to 40 mL 0.9% saline and pumped intravenously using a micropump at 7 mL per hour), the patient’s bilateral pupils were large and round, with a diameter of 3 mm. They were directly and indirectly sensitive to light reflection. The breath sounds of both lungs were rough, and obvious wet rales could be heard at the bottom of both lungs. Moreover, no murmurs in either valve area could be heard. His abdomen was flat, and the liver subcostal was not palpable. No edema in his lower limbs was observed. His Glasgow Coma Score was approximately three. His laboratory tests on 27 July showed the following results: routine blood test showed a white blood cell count of 22.73 × 10^9^/L; neutrophil ratio, 78.20%; red blood cells, 3.98 × 10^12^/L; platelets, 186 × 109/L; coagulation function test results showed plasma prothrombin time to be 18.60 s; plasma prothrombin standard ratio, 1.53; activated partial thrombin time, 39.40 s; blood biochemistry showed glutamic-pyruvic transaminase, 1648 IU/mL; glutamic oxaloacetic transaminase, 609 IU/mL; creatinine kinase, 278 IU/mL; creatine kinase isoenzyme, 4.40 ng/mL; total bilirubin, 15 μmol/L; and creatinine, 77 μmol/L. Laboratory results showed that the patient was in critical condition, with inflammatory indicators far beyond normal values, liver indicators seriously exceeding the standard, and abnormal clotting function. The ECG result is shown in Figure 2.

After admission, mechanical ventilation and bedside hemofiltration (CVVH mode) were maintained, and a combined treatment, including high-dose methylprednisolone, midazolam sedation (50 mg midazolam was added to 40 mL 0.9% normal saline, pumped via a micropump at 3 mL per hour), anti-infective treatment, and nutritional support, was administered. Laboratory examination on 29 July showed the following results: routine blood tests showed a white blood cell count of 19.96 × 10^9^/L; neutrophils, 87.20%; red blood cells, 4.21 × 10^12^/L; platelet count, 247 × 10^9^/L; blood biochemistry showed alanine aminotransferase, 591 IU/mL; aspartate aminotransferase, 124 IU/mL; creatine kinase, 1496 IU/mL; creatine kinase isoenzyme, 5.1 ng/mL; total bilirubin, 17.3 μmol/L; and creatinine, 65 μmol/L. The laboratory examination performed on the fourth day of admission demonstrated a serum myoglobin of 821.70 ng/mL; creatine kinase isoenzyme, 10.20 ng/mL; serum high-sensitivity troponin, 14,690.10 ng/L; and NT-probNP, 2811.00 pg/mL.

On 1 August 2018, the patient experienced a spike in temperature reaching 38.1 °C. The roxifen dosage (ceftriaxone sodium for injection) was then adjusted, and the fever resolved. The ECG performed on 5 August showed the following readings: 1. sinus rhythm; 2. moderate ST depression (ⅰ, ⅱ, avF, V3, V4, V5, and V6); and 3. abnormal T waves (ⅰ, ⅱ, avF, V3, V4, V5, and V6) (Figure 3). These findings were suggestive of anterior and inferior wall ischemia. On 14 August, the patient’s respiratory rate stabilized at 12–14 times/min. After intermittent weaning, the pharyngeal reflex was present; therefore, extubation was performed. Thereafter, the patient experienced no airway obstruction and his breathing stabilized.

On 15 August, the patient started coughing up yellow phlegm and had another fever spike; the highest reading reaching 38.9 °C. Sputum cultures and chest computed tomography examination were performed (Figure 4). The culture test results showed *Pseudomonas aeruginosa* and *Escherichia coli* growth. Consequently, the drug regimen was adjusted to imipenem 500 mg IV infusion Q 6 h according to the drug sensitivity test. The CT results showed increased texture of both lungs, a patchy high-density shadow, unobstructed bronchi above the segment, and a patchy high-density shadow of the lower and posterior segment of the right lung. Other laboratory tests revealed white blood cells, 7.12 × 10^9^/L; neutrophils ratio, 67.10%; red blood cells, 3.92 × 10^12^/L; platelets, 248 × 10^9^/L; serum myoglobin, 111.60 ng/mL; creatine kinase isoenzyme, 7.30 ng/mL; serum hypersensitive troponin I, 356.79 ng/L; and NT-probNP, 594.70 pg/mL. (Table 1) The patient’s symptoms significantly improved after weaning and adjusting antibiotics, and his body temperature gradually normalized. On 20 August, drug administration was stopped, and the patient’s condition improved; therefore, he was discharged. After discharge, he continued to take Jinshuibao and Qishen Yiqi pills orally.

On 1 September, the patient came back to the hospital for routine blood examination as well as liver and kidney function and myocardial damage marker tests. The lung CT reexamination revealed that the thorax was symmetrical, the bilateral lung textures had increased, nodules were observed in the middle of the right lung, and a small amount of high-density shadow was observed in the lower right lung, which was significantly smaller than before. The examination results of the patient had normalized and reached the standard for a clinical cure. Before the submission of this paper, we made a telephone appointment with the patient; the patient was in a complete recovery state, all the examinations were normal, and he could continue to engage in various labor activities.

## 3. Discussion

Aluminum phosphide is a type of highly effective, but toxic, insecticide for granary fumigation, known as ”wheat fumigation medicine”. Aluminum phosphide poisoning occurs via two routes, inhalation of phosphine gas through the respiratory tract or oral consumption of AP. Phosphine is a systemic agent; therefore, the toxic effects of both routes of poisoning are similar. However, as a highly toxic gas, phosphine inhalation through the respiratory tract has a stimulating effect on the respiratory tract when the human body interacts with a phosphine concentration greater than 9.7 mg/m^3^, which can cause poisoning with mild or moderate disturbance of consciousness, acute tracheo-bronchitis or peribronchial inflammation. At 550–830 mg/m^3^, 0.5–1 h exposure can lead to death [4] because phosphine extensively damages all organ systems [5], especially the circulatory and nervous systems, and leads to cellular metabolic disorders [6] that can manifest as refractory hypotension or shock, arrhythmias, and increased myocardial enzymes profile; liver damage, including jaundice and elevated aminotransferase; and disturbance of electrolyte and sugar metabolism. In severe cases, it can lead to multiple organ failure and death [7,8]. In the case we reported, the patient was transferred to our hospital in critical condition, required assisted ventilation using a ventilator, and developed aspiration pneumonia, acute left heart failure, and grade IV cardiac function. After active treatment, he finally achieved clinical resolution. There is no specific antidote for AP poisoning; therefore, comprehensive treatment is emphasized [9]. The patient should be removed from the poisoning site within a short time, transferred to a location with fresh air, and their airway kept unblocked. When dyspnea occurs, early application of ventilation to correct hypoxemia is crucial. Moreover, prevention and treatment of refractory hypotension and arrhythmias are the key determinants of a successful treatment. Timely restricted fluid resuscitation, glucocorticoids, vasoactive drugs, and bedside hemofiltration technology can be performed simultaneously to prevent and correct shock early on. Hemoperfusion can adsorb poisons and drugs dissolved in the blood to solid substances with rich surface area, and directly eliminates poisons and drugs from the blood, so as to rapidly reduce the concentration of poisons or drugs in the blood and organs, and interrupts the continued uptake of poisons or drugs by the organs and tissues. Glucocorticoids can effectively inhibit the release of inflammatory mediators and cytokines, stabilize cell membranes, and reduce inflammatory damage. During treatment, liver and kidney functioning should be protected; water status, electrolytes, and acid–base balance should be paid attention to; and hypokalemia should be especially prevented and treated [10,11].

AP poisoning often leads to fatal consequences. Circulatory failure and severe hypotension are common poisoning characteristics and common causes of death. Severe poisoning can also lead to multiple organ failure. With only supportive measures available, there is currently no specific antidote for AP poisoning. Early mechanical ventilation is one of the key treatment measures. AP poisoning can lead to intracellular respiratory depression, and effective oxygen supply is one of the keys to successful treatment. Early and sufficient short-term administration of adrenal glucocorticoid can significantly reduce the stress response of patients and alleviate the condition of pulmonary and cerebral edema. For those who are poisoned and have aspiration pneumonia, airway clearing and anti-infection therapy should be actively administered during treatment. At the same time, sedation, diuresis, gastrointestinal protection and nutritional nerve treatment should be administered according to changes in the patient’s condition. The findings of our case suggest that timely management with compliance to strict/stringent protocols that include certain operating specifications ensures positive outcomes, and that while using this chemical rationally, proper ventilation must be ensured to avoid inadvertent/accidental inhalation or consumption [12]. AP users can take some precautions at home. For example, to avoid children’s exposure to chemicals and avoid accidental ingestion, mark the cans/bottles for easy identification. These instructions must be posted by the manufacturer on bottles containing these chemicals. The early symptoms of aluminum phosphide poisoning are usually atypical, with gastrointestinal symptoms such as nausea and vomiting. If family members cannot provide a toxic exposure history, AP poisoning is easily misdiagnosed as acute gastroenteritis, myocarditis, etc., resulting in difficulties in diagnosis and even a delay in the best rescue time, resulting in major medical disputes. Therefore, publicity and education should be strengthened to make the public understand the toxicity of aluminum phosphide and related prevention measures, so as to avoid the occurrence of mass poisoning via aspiration and accidental ingestion.

This report has certain clinical significance for the treatment of acute aluminum phosphide poisoning, suggesting that doctors should closely observe changes in the disease during the course of treatment, to ensure the timely detection of the indexes of organs that may be damaged. At the same time, during the process of patient rescue, the authors found that the early use of short-acting glucocorticoids and the comprehensive application of bedside hemoperfusion technology have certain reference significance for improving the prognosis of patients and a good targeted guidance for clinical treatment.

## 4. Conclusions

In conclusion, improper use of aluminum phosphide can cause human poisoning, causing aspiration pneumonia, acute heart failure and other situations. At present, there is no specific antidote for phosphine poisoning. The comprehensive application of restricted fluid resuscitation, glucocorticoids, vasoactive drugs and bedside hemofiltration is of great significance to improve the prognosis of patients.

## Figures and Tables

**Figure 1 ijerph-20-05021-f001:**
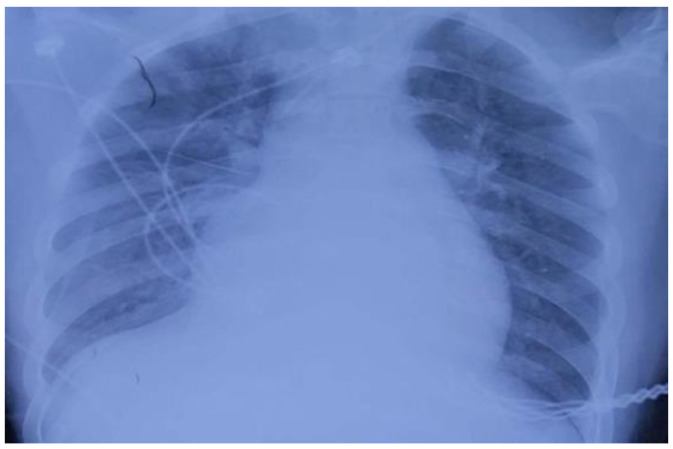
Bedside chest radiographs performed on 24 July 2018, showing extensive ground-glass changes in the right lung, exudate changes in both lungs, and an enlarged heart boundary.

**Figure 2 ijerph-20-05021-f002:**
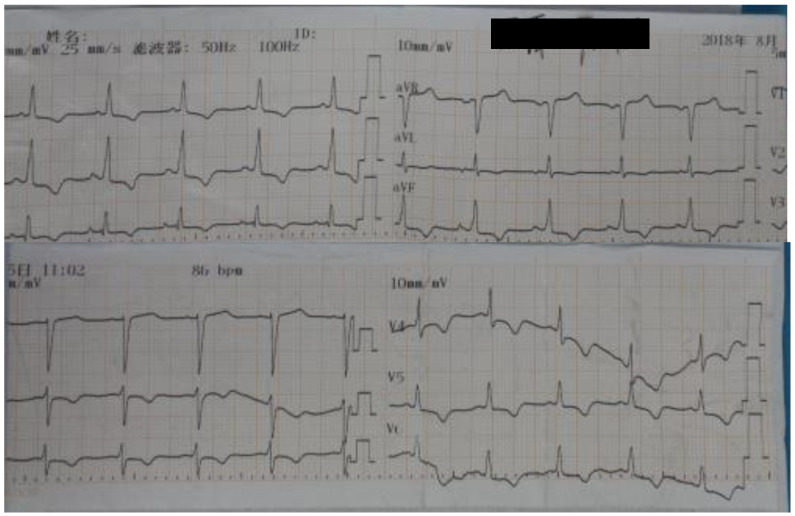
Electrocardiogram performed on 5 August 2018 showing: 1. sinus rhythm, 2. moderate ST depressions (ⅰ, ⅱ, avF, V3, V4, V5, and V6), and 3. abnormal T waves (ⅰ, ⅱ, avF, V3, V4, V5, and V6), suggestive of anterior and inferior wall ischemia.

**Figure 3 ijerph-20-05021-f003:**
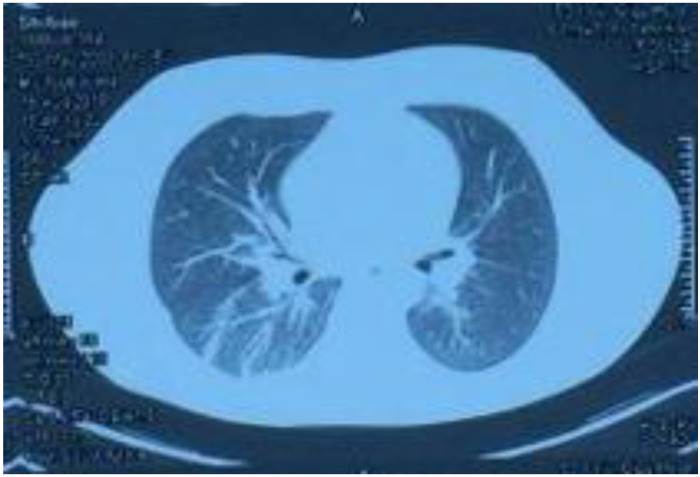
Lung CT performed on 15 August 2018, showing increased texture in both lungs, a patchy high-density shadow, unobstructed bronchi above the segment, and a patchy high-density shadow in the posterior segment of the lower lobe of the right lung.

**Figure 4 ijerph-20-05021-f004:**
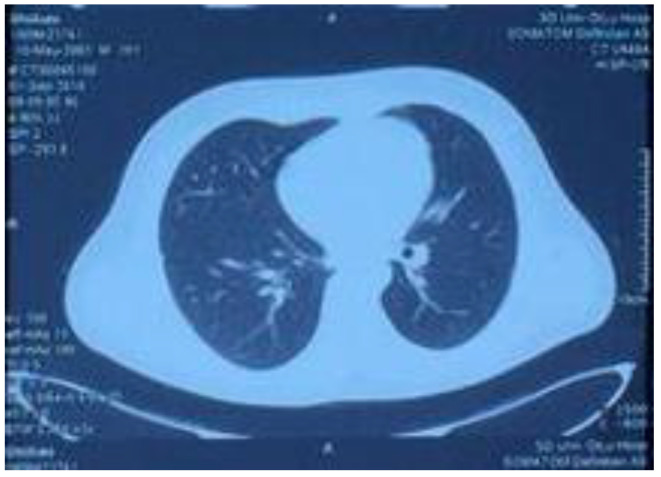
Lung CT reexamination performed on 1 September 2018, showing symmetrical thorax, increased bilateral lung textures, nodules in the middle lobe of the right lung, and a small amount of high-density shadow in the lower lobe of the right lung, which was significantly smaller than before.

**Table 1 ijerph-20-05021-t001:** Timeline with Relevant Data from the Episode of Care.

	7.25	7.26	7.27	7.29	8.9	8.15
ALT IU/mL	2934	2536	1648	591	90	69
AST IU/mL	1870	857	609	124	57	47
TBIL μmol/L	30.7	30.7	15	17.3	8.3	11
IBIL μmol/L	25.7	-	7	9.7	4.2	7
Cr μmol/L	-	58.4	77	65	34	44
WBC × 10^9^/L	-	24.79	22.73	19.96	17.77	7.12
NEU%	-	19.90	78.20	87.20	84.50	67.10
RBC × 10^12^/L	-	-	3.98	4.21	4.44	3.92
PLT × 10^9^/L	-	-	186	247	253	248
PT-S s	-	18.2	18.60	19.2	13.3	14.0
PT-INR s	-	1.37	1.53	1.70	1.20	1.07
APTT	-	-	39.40	34.4	34.10	41.50
CK IU/mL	-	344	278	1496	693	111.60
CK-MB ng/mL	-	10	4.40	5.1	33.1	7.30

ALT: alanine aminotransferase; AST: aspartate aminotransferase; TBIL: total bilirubin; IBIL: inter bilirubin; Cr: creatinine; WBC: white blood cells; NEU%: neutrophils; RBC: red blood cell; PLT: platelet; PT-S: plasma prothrombin time; PT-INR: plasma prothrombin standard ratio; APTT: activated partial thromboplastin time; CK: creatine kinase; CK-MB: creatine kinase isoenzyme. The tests were conducted between 25 July and 15 August.

## Data Availability

The original contributions presented in the study are included in the article, further inquiries can be directed to the corresponding author/s.

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
