# Peer review of "Case Report: Acute Intoxication from Phosphine Inhalation"

_ijerph, 2023, doi:10.3390/ijerph20065021_

Round 1
Reviewer 1 Report
Line 16: Replace “were” with “was” after “survey”
Line 17: Remove “s” from “patients”
Line 256-27: Remove one of the two “highly toxic” in these two lines to avoid redundancy
Line 34: Add location of the hospital
Line 134: Expand “AP”
Add more information about the granary where this incidence happened? How far it was from the local hospital where the patient was admitted on July 23rd? Timings of the incidence and hospital admission are also needed.
Add patient’s age, weight and GCS score when he entered into a coma
What could be the reason behind experiencing nausea and chest-tightness by the patient on the day before the actual poisoning was reported? Were all tests normal?
Are the treatments suggested in this report equally effective for all age groups?
Reviewer 2 Report
This case report is very useful and can be served as reference for clinical practise. Some of the comments are as follows.
Line 16 – Typo. Please change “were” to “was”.
Line 16-17 – Please provide the duration of the patient for life support treatment.
Section 2 – Additional information of the patient are required. These include his age, gender, weight/height and underlying diseases.
Line 65-71 – Authors should briefly discuss if the results were considered bad or severe.
Figure 2 – The image’s resolution is too poor. It can’t be clearly read.
Figure 3 and 4 – I strongly advised the authors to combine both images and arrange side-by-side in order to make comparison easier.
Section 3 – A text description should be provided for the table. Besides, please remove all arrow symbols from the table. Also, what does “7.25 – 8.15” mean for?
Lastly, if possible, please provide an update on the current medical condition of the patient mentioned in this paper. Any side-effect after the poisoning.
Reviewer 3 Report
In this case report, the authors share important and practical knowledge regarding how to deal with a intoxicated patient after phosphine inhalation. The manuscript is generally well-written and comprehensive and authors should receive credit for sharing this knowledge and experience with colleagues that might experience similar clinical cases in the future. However, I believe some aspects need to be addressed before publication.
1) The abstract structure does not follow the instructions provided in "Intructions for authors". Here are the instructions provided: "The abstract should be a single paragraph and should follow the style of structured abstracts, but without headings: 1) Background: Place the question addressed in a broad context and highlight the purpose of the study; 2) Methods: Describe briefly the main methods or treatments applied. Include any relevant preregistration numbers, and species and strains of any animals used. 3) Results: Summarize the article's main findings; and 4) Conclusion: Indicate the main conclusions or interpretations. The abstract should be an objective representation of the article: it must not contain results which are not presented and substantiated in the main text and should not exaggerate the main conclusions."
2) This is the major point that I think must be addressed. I belive the introduction should be brief (as it is) in a case report. However, according to the instructions it should also provide information regarding " general medical condition or relevant symptoms that will be discussed in the case report". Thus, I advice the authors to provide information regarding this as well. International guidelines regarding how to manage patients after phosphine inhalation are provided by international entities. Therefore, how is your case report important to be published? How is it different form other patients/treatment recommendations? In other words, why should readers read your case report in stead of/in addition to read these guidelines?
3) Line 44-45: Why is this sentence in italics?
4) Line 79: I do not understand why this concepts are underlined.
5) Figure 2 - showing the electrocardiogram of the patient is important in this case report. However, it is difficult for the reader to analyse it with detail and understand what authors highlight in the figure caption. If possible, the image should have a better quality or be bigger.
6) "While in the sedated state, the patient’s bilateral pupils were large and round, with a diameter of 3 mm". So you sedated this patient, right? At least during some parts of the recovery. I believe it would be important to mention which drug and dose did you used to sedate this patient. Case reports should also provide very practical data and feedback that can be usefull to other colleagues. Thus, I believe the durgs used and the dosis should always be provided.
I have nothing further to add.
